# Identification and Comparison of microRNAs in the Gonad of the Yellowfin Seabream (*Acanthopagrus Latus*)

**DOI:** 10.3390/ijms21165690

**Published:** 2020-08-08

**Authors:** Shizhu Li, Genmei Lin, Wenyu Fang, Dong Gao, Jing Huang, Jingui Xie, Jianguo Lu

**Affiliations:** 1School of Marine Sciences, Sun Yat-sen University, Zhuhai 519082, China; lishizhu@mail.sysu.edu.cn (S.L.); lingm5@mail.sysu.edu.cn (G.L.); fangwy5@mail2.sysu.edu.cn (W.F.); 624618874@163.com (D.G.); huangj393@mail2.sysu.edu.cn (J.H.); xiejg5@mail2.sysu.edu.cn (J.X.); 2Southern Marine Sciences and Engineering Guangdong Laboratory, Zhuhai 519082, China

**Keywords:** yellowfin seabream, small RNA sequencing, sex-biased miRNA, sexual differentiation, gonadal development

## Abstract

Yellowfin seabream (*Acanthopagrus latus*) is a commercially important fish in Asian coastal waters. Although natural sex reversal has been described in yellowfin seabream, the mechanisms underlying sexual differentiation and gonadal development in this species remain unclear. MicroRNAs (miRNAs) have been shown to play crucial roles in gametogenesis and gonadal development. Here, two libraries of small RNAs, constructed from the testes and ovaries of yellowfin seabream, were sequenced. Across both gonads, we identified 324 conserved miRNAs and 92 novel miRNAs: 67 ovary-biased miRNAs, including the miR-200 families, the miR-29 families, miR-21, and miR-725; and 88 testis-biased miRNAs, including the let-7 families, the miR-10 families, miR-7, miR-9, and miR-202-3p. GO (Gene Ontology) annotations and KEGG (Kyoto Encyclopedia of Genes and Genomes) enrichment analyses of putative target genes indicated that many target genes were significantly enriched in the steroid biosynthesis pathway and in the reproductive process. Our integrated miRNA-mRNA analysis demonstrated a putative negatively correlated expression pattern in yellowfin seabream gonads. This study profiled the expression patterns of sex-biased miRNAs in yellowfin seabream gonads, and provided important molecular resources that will help to clarify the miRNA-mediated post-transcriptional regulation of sexual differentiation and gonadal development in this species.

## 1. Introduction

MicroRNAs (miRNAs) are small, endogenous, non-coding RNAs that are approximately 22 nucleotides long; miRNAs regulate gene silencing and translational repression by binding to target messenger RNAs (mRNAs) [1]. The relationships between miRNAs and mRNAs are intricate, in that one miRNA may regulate several genes, and each gene may also be targeted by multiple miRNAs [2]. Since miRNAs were first discovered in 1993, thousands have been identified in diverse animal groups, including mammals, birds, amphibians, and fish, and it has been shown that most mature miRNA sequences are conserved across these groups [3,4,5,6]. MiRNAs play important roles in the regulation of multiple biological processes, such as cell proliferation and differentiation, embryonic development, apoptosis, energy metabolism, immunity, metamorphosis, and gametogenesis [7,8,9,10,11,12,13].

Sexual development is a complex process that is tightly regulated by a series of molecules and signaling networks at transcriptional and post-transcriptional levels [14]. As high-throughput sequencing technologies have improved, small RNA transcriptome sequencing has become commonly used to study the expression profiles of miRNAs, and to identify the corresponding target genes. Several recent studies have reported that miRNAs are differentially expressed during sexual differentiation and gonadal development. Dynamic miRNA expression patterns in primordial germ cells and gonadal somatic cells at different embryonic stages have been shown in mice [15]. In addition, it was shown that miRNAs were differentially expressed between sexually mature and immature ovaries in chickens [16]. In zebrafish, the miRNA repertoires were characterized in the undifferentiated state at three weeks post fertilization (wpf) to mature adults at 24 wpf [17]. Moreover, many sex-biased miRNAs have been identified in several aquaculture species. For example, in yellow catfish, some miRNAs (i.e., miR-21-5p, miR-21-3p, and miR-462-5p) were strongly expressed in the ovary, while others (e.g., miR-9-3p, miR-103b-3p, and miR-7b) were mainly expressed in the testis [18,19]. In Nile tilapia, the miRNA expression profiles were analyzed in the gonads from sexual differentiation to sexual maturity; the miR-727, miR-129, and miR-29 families were highly expressed in the mature ovary, whereas the miR-132, miR-212, miR-33a, and miR-135b families were significantly more highly expressed in the sexually mature testis [20]. Additionally, some miRNAs, including miR-34c-5p, miR-153-5p, and miR-749, were expressed exclusively in Nile tilapia XX gonads at five days post-hatching, whereas miR-1306-5p, miR-132b, and miR-18c were expressed exclusively in the XY gonads [21]. MiRNA expression patterns have also been studied in the gonads of *Oplegnathus punctatus* [22], *Scylla paramamosain* [23,24], *Acipenser schrenckii* [25], *Trachinotus ovatus* [26], and *Odontobutis potamophila* [27]. These studies help to clarify the roles of miRNAs in sexual differentiation and gonadal development.

Yellowfin seabream (*Acanthopagrus latus*), a fish found in Asian coastal waters, is commercially important because of its appearance, taste, and nutritional value [28]. Apart from its economic value, yellowfin seabream is a good model for studies of sexual differentiation because of its reproductive biology: this fish develops as a functional male during its first reproductive cycle, then undergoes a long-term sexual reversal and becomes female [29]. However, limited genomic resources are available for the yellowfin seabream. Consequently, it is important to understand the mechanisms of sexual differentiation and gonadal development at the genomic level in this species. Here, small RNA transcriptome sequencing technology was used to identify known and novel miRNAs in the yellowfin seabream and to explore the miRNA networks underlying the post-transcriptional regulation of sex differentiation. Sex-biased miRNAs and their integrations with potential targets were analyzed to clarify their interaction networks. Our findings will help to clarify the potential roles played by miRNAs in yellowfin seabream sexual differentiation and gonadal development.

## 2. Results

### 2.1. Overview of the Small-RNA Sequencing Data

We generated 14,286,897 and 10,581,727 raw reads from the testes and ovaries, respectively. After trimming adapter sequences, removing low-quality reads, and discarding sequences of inappropriate length (i.e., exceeding the range 18–32 bp), 12,499,328 clean reads from the testes and 8,188,452 clean reads from the ovaries remained for further analysis (Table A2). The length distributions of the clean reads differed between the two libraries. In the ovary, there were two peaks: one at 22 nt representing typical Dicer-derived products, and one at 28 nt representing the piwi-interacting RNAs (piRNAs). In contrast, there was only one piRNA peak in the testis (Figure 1).

### 2.2. Identification of Conserved and Novel miRNAs in the Gonads of the Yellowfin Seabream

By searching miRBase 22.0 and predicting novel miRNAs with miRDeep2, 324 known miRNAs and 92 novel miRNAs were identified in the yellowfin seabream gonads. Of these, 37 known miRNAs and 26 novel miRNAs were specific to the testis, whereas 51 known miRNAs and 5 novel miRNAs were specific to the ovary. The remaining 236 known miRNAs and 61 novel miRNAs were expressed in both the testis and the ovary (Figure 2A,B). In total, 360 mature miRNAs were identified in the testis, and 353 mature miRNAs were identified in the ovary. Other non-coding RNAs identified in the yellowfin seabream gonads, including rRNAs, tRNAs, snRNAs, snoRNAs, and repeat sequences, are shown in Table A3.

### 2.3. MiRNA Expression Patterns in Yellowfin Seabream Gonads

Of the 360 mature miRNAs identified in the testis, 95 miRNAs (26.4%) were expressed at low levels (0 < TPM (transcripts per million) < 10), 183 miRNAs (50.8%) were expressed at moderate levels (10 < TPM < 1000), and 82 miRNAs (22.8%) were expressed at high levels (TPM > 1000). In comparison, of the 353 mature miRNAs identified in the ovary, 114 miRNAs (32.3%) were expressed at low levels (0 < TPM < 10), 168 miRNAs (47.6%) were expressed at moderate levels (10 < TPM < 1000), and 71 miRNAs (20.1%) were expressed at high levels (TPM > 1000) (Figure 2C,D). No obvious differences in the distributions of miRNA expression levels were observed between the testis and the ovary.

The 10 most abundant miRNAs in the testis were miR-143, miR-30d, miR-15b, and seven novel miRNAs, whereas the 10 most abundant miRNAs in the ovary were miR-143, miR-30d, miR-181c, and 7 novel miRNAs. Of these, miR-143, miR-30d, miR-novel-46, miR-novel-47, and miR-novel-70 were highly abundant in both the testis and the ovary of the yellowfin seabream (Table 1).

### 2.4. Identification of Sex-Biased miRNAs in the Testis and the Ovary

We identified 155 miRNAs that were differentially expressed between the testes and the ovaries of yellowfin seabream (Figure 3): 88 testis-biased miRNAs (41 known miRNAs and 47 novel miRNAs) and 67 ovary-biased miRNAs (55 known miRNAs and 12 novel miRNAs). Of these, 24 miRNAs were exclusively expressed in the testes and 6 miRNAs were specifically expressed in the ovaries. Excluding these gonad-specific miRNAs, the greatest fold-change in expression level between the testis and the ovary was a 92.5-fold change in testis-biased miRNA expression and a 38.7-fold change in ovary-biased miRNA expression. In addition, the expression levels of 20 testis-biased miRNAs (31.25%) were over 8-fold greater than those in the ovary, whereas the expression levels of 10 ovary-biased miRNAs (16.39%) were over 8-fold greater than those in the testis. Thus, more testis-biased miRNAs were differentially expressed than ovary-biased miRNAs, and the fold-changes in expression levels were greater. Overall, the let-7 families, the miR-10 families, miR-7, miR-9, and miR-202-3p were highly expressed in the testis, whereas the miR-200 families, the miR-29 families, miR-21, and miR-725 were upregulated in the ovary. The 30 most DE miRNAs between the testis and the ovary are shown in Figure 4.

### 2.5. Validation of miRNA Expression Using RT-qPCR

To validate the transcriptome data, we selected five testis-biased miRNAs (miR-724, miR-7a, miR-202-3p, miR-novel-5, and miR-novel-49) and five ovary-biased miRNAs (miR-725, miR-216b, miR-200a, miR-novel-35, and miR-novel-69) for RT-qPCR analysis. The RT-qPCR results indicated that miR-724, miR-7a, miR-202-3p, miR-novel-5, and miR-novel-49 were more highly expressed in the testis than in the ovary. Conversely, miR-725, miR-216b, miR-200a, miR-novel-35, and miR-novel-69 were more highly expressed in the ovary than in the testis (Figure 5). These results were consistent with our small RNA-seq results, indicating that our sequencing results were reliable.

### 2.6. Target Gene Prediction and Functional Analysis

To investigate the potential functions of the DE miRNAs, we predicted the corresponding target genes. Our analyses predicted 16,217 target genes of the DE miRNAs, which were classified into 57 GO (Gene Ontology) subcategories: 26 biological process terms, 16 cellular component terms, and 15 molecular function terms. Of the biological process terms, the target genes were mainly associated with cellular process (GO: 0009987), single-organism process (GO: 0044699), and metabolic process (GO: 0008152). Of the cellular component terms, the target genes were mainly associated with membrane (GO: 0016020), membrane part (GO: 0044425), and cell (GO: 0005623). Of the molecular function terms, the target genes were mainly associated with binding (GO: 0005488), catalytic activity (GO: 0003824), and transporter activity (GO: 0005215) (Figure 6). KEGG (Kyoto Encyclopedia of Genes and Genomes) pathway enrichment analysis indicated that the target genes were annotated in 131 signaling pathways. The 20 most enriched KEGG pathways are shown in Figure 7. Notably, eight pathways were significantly enriched (*p* < 0.05), including the phosphatidylinositol signaling system (ko04070), endocytosis (ko04144), and steroid biosynthesis (ko00100).

Next, we analyzed several important GO terms and KEGG pathways associated with sexual development and reproduction enriched in the putative target genes, including the GO term “reproductive process” and the steroid biosynthesis signaling pathway. We identified 61 potential target genes associated with reproductive processes, including zona pellucida glycoprotein 3 (*zp3*), sperm acrosome membrane-associated protein 4 (*spaca4*), annexin A1 (*anxa1*), mitotic-specific cyclin-B1 (*ccnb1*), and meiotic recombination protein *dmc1*. In addition, 17 putative target genes were enriched in the steroid biosynthesis signaling pathway, including 17β-hydroxysteroid dehydrogenase type 7 (*hsd17b7*), cytochrome P450 family 24 subfamily a1 (*cyp24a1*), and squalene monooxygenase (*sqle*).

### 2.7. Integrated Analysis of the DE miRNAs and Corresponding Target Genes

To understand the relationships between DE miRNAs and the corresponding sex-related target genes, we constructed regulatory networks (Figure 8). MiR-202-3p was predicted to target *spaca4*, *anxa1*, *ccnb1*, *sqle*, and sperm flagellar protein 1; these genes are associated with gametogenesis, cellular signaling, and hormonal secretion [30,31,32,33,34]. The testis-biased miRNA let-7j was predicted to target *cyp3c* and *lhb*, two ovary-biased genes that may be associated with steroids synthesis and the sex hormone responsible for follicular development [35]. The ovary-biased miR-29b might target *cyp11b1*, which encodes a rate-limiting enzyme of steroid hormones synthesis that is required for testis development [36]. MiR-29b may also target meiotic recombination protein *dmc1*, which is essential for spermatogenesis [37].

Previously, we have identified many sex-biased genes through the transcriptome sequencing of yellowfin seabream gonads [38]. Here, the interaction networks of the sex-related genes and the DE miRNAs were analyzed (Figure 8). Because miRNAs interact with coding genes to negatively regulate expression, most pairs of interacting miRNAs and mRNAs exhibited negatively correlated expression patterns in the gonads. The *sox10* gene, which is associated with testicular differentiation, was targeted by 15 miRNAs, nine of which were ovary-biased (miR-222a-3p, miR-221-3p, miR-150, miR-181a, miR-212b, miR-210-5p, miR-10544, miR-novel-56, and miR-novel-17). Similarly, the testis-determining gene *sox9* was targeted by the ovary-biased miRNAs miR-124, miR-novel-86, and miR-novel-87. *Foxh1*, a factor required for establishment of oocyte polarity, was predicted to be a target of three testis-biased miRNAs (miR-103, miR-107, and miR-novel-49). *Dax1*, a key transcription factor in regulation of ovarian differentiation, was potentially targeted by eight testis-biased miRNAs (miR-1306, miR-727, miR-27b-5p, miR-novel-49, miR-novel-66, miR-novel-62, miR-novel-9, and miR-novel-83). These results suggested that sexual differentiation and gonadal development might be regulated by an intricate miRNA-mRNA network in yellowfin seabream.

## 3. Discussion

Recently, miRNAs have been recognized as key post-transcriptional regulators of gene expression in many biological processes [8,10,39,40,41]. The important role of miRNAs in sexual differentiation and gonadal development has been reported in diverse animals, including mammals, birds, and fish [15,16,17]. Small RNA high-throughput sequencing has identified sex-related miRNAs in a wide range of fish species, including *Danio rerio* [17,42], *Pelteobagrus fulvidraco* [18,19], *Oreochromis niloticus* [20,21], *Oplegnathus punctatus* [22], *Acipenser schrenckii* [25], *Trachinotus ovatus* [26], and *Odontobutis potamophila* [27]. Here, miRNA expression profiles and interaction networks were investigated in yellowfin seabream. These results will help to clarify the mechanisms underlying sexual differentiation in this species.

The overall distribution of small RNAs differed between the testis and the ovary. We observed both miRNA and piRNA peaks in the ovary, but only one piRNA peak in the testis. PiRNAs were more plentiful in the testis, whereas miRNAs were more enriched in the ovary. The most abundant small RNAs in the gonads were the piRNAs, which play critical roles in gametogenesis, especially testis development [43,44]. During the long term of oogenesis, many materials, including mRNAs, proteins, and miRNAs, are stored in the egg, as preparation for early embryonic development.

To better compare the similarities and differences of the miRNA expression profiles in *Acanthopagrus latus* and six reported fishes. Except for the species-specific miRNAs, we summarized nine testis- or ovary-biased miRNAs, including let-7, miR-10, miR-7, miR-9, miR-202, miR-200, miR-29, miR-21, and miR-725, which exhibited significant sex-biased expression in at least four fish species (Table 2) [18,19,20,21,22,25,26,27]. MiR-7 exhibited a significant testis-biased expression in *Acanthopagrus latus* and three other fish species, indicating that it might play an important role in testis development. However, the other eight miRNAs exhibited different gender biases in different fish species. For example, let-7 was highly expressed in the testis of *Acanthopagrus latus*, *Pelteobagrus fulvidraco, Acipenser schrenckii*, and *Odontobutis potamophila,* whereas it was identified as an ovary-biased miRNA in *Oplegnathus punctatus*; miR-9 was a testis-biased miRNA in *Acanthopagrus latus*, *Pelteobagrus fulvidraco, Acipenser schrenckii*, and *Trachinotus ovatus*, but was mainly expressed in *Oreochromis niloticus.* This result indicates that they may play different roles in gonadal development in different species.

It has been reported that miR-143 is highly expressed during gonadal development, and plays an important role in this process. For example, miR-143 inhibits steroid hormone synthesis and granulosa cell apoptosis by targeting FSHR (follicle stimulating hormone receptor) in the bovine ovary [45]. In mice, miR-143 regulates granulosa cell proliferation by targeting the FSH signaling pathway [46]. The miR-30 family was highly expressed in the testes of *Trachinotus ovatus* [26], yellow catfish [19], and Amur sturgeon [19,25]. The miR-181 family is highly expressed in the gonads of tilapia [21], mice [47], and humans [48]. In yellowfin seabream gonads, miR-143, miR-30d, and miR-181c were the most highly expressed miRNAs, indicating that these might participate in gonadal development in the yellowfin seabream, similar to what has previously been shown in other animals.

In mice, the let-7 family was expressed in the spermatogonia and spermatocytes, and was associated with spermatogonial differentiation through the retinoic acid signaling pathway [49]. The let-7 miRNA families were also identified as testis-biased in Amur sturgeon [25], Atlantic halibut [50], and yellow catfish [19]. However, the expression patterns of the let-7 families differed in spotted knifejaw, where the let-7 families were more highly expressed in the ovary than in the testis [22]. In mice, miR-7a2 affected sexual maturation and reproductive processes by regulating the synthesis and secretion of gonadotropic and sex-steroid hormones; notably, the deletion of miR-7a2 caused infertility in both sexes [51]. MiR-7 was also highly expressed in yellow catfish [18], suggesting that this miRNA might participate in testis development in fish. It has been suggested that MiR-9 might regulate sexual development in several species. For example, miR-9 was upregulated in the tilapia ovary and was predicted to affect sexual determination and differentiation by regulating the expression of *dmrt1* [20]. In contrast, in *Monopterus albus*, miR-9 might affect the physiological processes that promote oocyte degeneration and might stimulate spermatogenesis via *foxl3* expression during natural sex change [52]. Here, the let-7 families, miR-7, and miR-9 exhibited testis-biased expression patterns, suggesting that these miRNAs might participate in spermatogenesis and testis development in yellowfin seabream.

Some miRNAs were also strongly expressed in the yellowfin seabream ovary, including the miR-200 family (miR-200a, miR-200b, and miR-429a), the miR-29 family (miR-29a, miR-29b, and miR-29c), and miR-21. In mice, miR-200b and miR-429 inhibited luteinizing hormone synthesis by negatively regulating *zeb1*, which is required for ovulation and female fertility [53]. In zebrafish, miR-200 clusters regulate sperm motility by targeting *amh*, *wt1a,* and *srd5a2b* [41]. The miR-200 family is also highly expressed in the ovaries of several other fish species [18,22]. In mice, the miR-29 family might participate in male meiosis by downregulating *Dnmt3a*, which plays an important role in the response to DNA damage [54], while miR-29-3p and its target *Tbx21* help to regulate the onset of puberty and reproduction in female mice by modulating *Gnrh1* expression [55]. In addition, miR-29 is an ovary-biased miRNA in tilapia [21]. MiR-21 is required to block the apoptosis of granulosa cells during mouse periovulation [56]. Moreover, miR-21 regulates ovarian development in rainbow trout [57]. Thus, the ovary-biased expression patterns of the miR-200 family, the miR-29 family, and miR-21 suggested that these miRNAs might be involved in ovarian development in yellowfin seabream.

As miRNAs themselves cannot encode functional protein products, these small RNAs affect biological function by the post-transcriptional regulation of target genes. The predicted target genes of the DE miRNAs were classified into 57 GO subcategories. Of these, 61 putative target genes were enriched in the GO term “reproductive process”, suggesting that the corresponding DE miRNAs were involved in yellowfin seabream reproduction. In addition, the putative target genes were significantly enriched in eight KEGG pathways, including “steroid biosynthesis”, “endocytosis”, and “phosphatidylinositol signaling system”. Sex steroids are tightly associated with sexual differentiation, gonadal development, and sexual reversal [58,59,60]. Endocytosis is required for the cellular uptake of sex steroids and yolk during oogenesis [61,62]. The phosphatidylinositol signaling system is primarily associated with the phosphatidylinositol-3 kinase (PI3K) pathway, which is critical for gonadal development [63]. Therefore, the DE miRNAs might critically affect gonadal development by targeting their gonad-development-associated target genes.

Each miRNA may interact with many different target genes, and each coding gene may be targeted by many miRNAs. MiR-202-3p was predicted to target *spaca4*, *anxa1*, *ccnb1*, *sqle*, and sperm flagellar protein 1. Conversely, *cyp11b1* was predicted to be the target of miR-29b, miR-181a, miR-199-3p, and miR-novel-64. Therefore, miRNAs and mRNAs may form complicated interaction networks. Here, we constructed integrated networks of the DE miRNAs and known genes associated with sexual differentiation. These networks demonstrated negatively correlated expression patterns: ovary-biased DE miRNAs always targeted genes associated with testis development, while the testis-biased DE miRNAs always targeted genes associated with ovarian development. The testis-determining gene *sox9* was targeted by three ovary-biased miRNAs (miR-124, miR-novel-86, and miR-novel-87), which suggests that inactivating *sox9* expression might potentially maintain ovarian development in yellowfin seabream. Indeed, it has been demonstrated that miR-124 prevents *sox9* expression during ovarian development [64]. Meanwhile, the oocyte-polarity marker gene *foxh1* was targeted by miR-103, miR-107a-3p, and miR-novel-49; three miRNAs that were significantly downregulated in the yellowfin seabream ovary. Therefore, repression of miR-103, miR-107a-3p, and miR-novel-49 might be required for oogenesis in yellowfin seabream. The miRNA-mRNA interaction network provides significant information for the identification of miRNAs associated with gonadal development. The direct interactions between miRNAs and their predicted targets, and its role in gonadal development will be analyzed in our future studies.

## 4. Materials and Methods

### 4.1. Ethical Procedures

All procedures with *Acanthopagrus latus* were approved by the Ethics Committee of Sun Yat-Sen University (protocol no. 20200110008, approval date: 11 October 2019) and the methods were carried out following the approved guidelines.

### 4.2. Sample Collection

Yellowfin seabreams were obtained from Hailv aquaculture station (Zhuhai, China). Six yellowfin seabreams were selected: three 1-year-old males, with an average body weight (ABW) of 113.3 ± 8.5 g and an average body length (ABL) of 15.3 ± 0.3 cm, and three 3-year-old females, with an ABW of 519.3 ± 55.3 g and an ABL of 26.1 ± 0.8 cm. Each fish was anesthetized using 100 mg/L tricaine methane sulfonate (MS-222). Fish gonads were extracted, snap frozen in liquid nitrogen, and stored at −80 °C for RNA extraction.

### 4.3. RNA Extraction, Library Construction, and Small RNA Sequencing

Samples were removed from the −80 °C freezer and homogenized for 40 seconds with Bioprep-24 (Tomos, Shanghai, China) in 1 mL TRIzol reagent (Invitrogen, Carlsbad, CA, USA). Total RNAs were extracted according to the manufacturer’s instructions. The extracted RNA was further treated with DNaseI (Vazyme, Nanjing, China) to eliminate genomic DNA contamination. RNA degradation and contamination were further monitored by 1.5% agarose gel. The quantity and quality of the isolated RNA were measured using a BioSpec-nano Spectrophotometer (Shimadzu, Japan), and the RNA integrities were assessed by Agilent 2100 Bioanalyzer (Agilent Technologies, Santa Clara, CA, USA). Only RNAs with an OD260/280 ≥ 1.8 and an RIN (RNA integrity number) ≥ 7 were retained. The total RNA of each sample was standardized to 200 ng/μL, then an equal amount of high-quality RNA (10 μL) from each sample was pooled (testes and ovaries were pooled separately), and sent to Majorbio Bio-pharm Technology Co., Ltd. (Shanghai, China) for small RNA library construction and sequencing. Briefly, RNA was purified by PAGE to enrich for 18–32 nt molecules, ligated with 5′- and 3′- adaptors, and was then sequenced using the Illumina HiSeq platform.

### 4.4. Quality Control and miRNA Identification 

After sequencing, the raw reads underwent a strict quality control routine to remove unreliable reads, including trimming adapter sequences, discarding low-quality reads (with Q-scores < 20 or containing poly-N), and eliminating reads shorter than 18 bp or longer than 32 bp. Because the yellowfin seabream lacks a reference genome, the clean reads were aligned to the genome of the black seabream (a species related to the yellowfin seabream; http://gigadb.org/dataset/100409) using Bowtie (http://bowtie-bio.sourceforge.net/index.shtml). Known miRNAs were identified by aligning the mapped reads to the mature miRNAs of *Neolamprologus brichardi*, *Pundamilia nyererei*, *Oreochromis niloticus*, and *Danio rerio* in miRBase 22.0 (http://www.mirbase.org/). The unannotated small RNA reads were compared with the Rfam database (http://rfam.xfam.org/) and RepeatMasker (http://www.repeatmasker.org/) to remove rRNAs (ribosomal RNA), tRNAs (transfer RNA), snRNAs (small nuclear RNA), snoRNAs (small nucleolar RNA), and repeat sequences. Novel miRNAs were predicted in the remaining reads using miRDeep2 (https://www.mdc-berlin.de/content/mirdeep2-documentation).

### 4.5. Differential miRNA Expression Analysis

MiRNA expression levels were standardized based on the number of transcripts per million reads (TPM). Differentially expressed miRNAs (DE miRNAs) between the testes and the ovaries were identified using DEGSeq Version 1.30.0 (http://bioconductor.org/packages/stats/bioc/DEGSeq/), correcting the false discovery rate with the Benjamini–Hochberg method. MiRNAs with an absolute fold-change value ≥ 2 and an adjusted *p*-value < 0.001 were identified as significant DE miRNAs.

### 4.6. Real-Time Quantitative PCR Verification (RT-qPCR)

To validate the transcriptome data, a total of 10 miRNAs were screened for RT-qPCR, including 5 testis-biased miRNAs and 5 ovary-biased miRNAs. The RNAs used for RT-qPCR verification were the same as that for transcriptome sequencing. Briefly, testis and ovary tissues were dissected from three one-year-old and three-year-old individuals, respectively. Total RNA was isolated from each sample using TRIzol reagent (Invitrogen, USA). The extracted RNA from three individuals in each group (testes and ovaries) were mixed in equal mass ratios, and then reverse transcribed using Mir-X miRNA First-Strand Synthesis Kits (Takara, Dalian, China), following the manufacturer’s instructions. Roche Light Cycler 480 (Roche, Switzerland) was applied to perform RT-qPCR with TB Green Premix Ex Taq (Takara, Dalian, China). The thermal cycling conditions were as follows: 94 °C for 5 min; 40 cycles of 94 °C for 12 s, 58 °C for 12 s, and 72 °C for 10 s; and an additional 72 °C for 2 min. After thermal cycling, a final disassociation curve analysis was performed. Each sample was analyzed in triplicate, and miRNA relative expression levels were normalized to those of U6 snRNA using the 2^−ΔΔCT^ method. Statistical analysis was performed by SPSS 20.0. Differences with *p* values < 0.05 were considered statistically significant. The primers used in this study are listed in Table A1.

### 4.7. Prediction of DE miRNA Targets

The potential target genes of the DE miRNAs were predicted using miRanda (http://www.miranda.org/), Targetscan (http://www.targetscan.org/), and RNAhybrid (http://bibiserv.techfak.uni-bielefeld.de/rnahybrid/). The 3′ UTRs (untranslated region) of all of the genes were extracted from the yellowfin seabream gonad transcriptome (SRA accession number: PRJNA622226), and matched to the miRNAs. Gene Ontology (GO) annotations and Kyoto Encyclopedia of Genes and Genomes (KEGG) enrichment analyses were performed to identify the functions and pathways associated with these candidate target genes. 

## 5. Conclusions

In summary, the small-RNA profiles of the yellowfin seabream testis and ovary were obtained. We identified 324 conserved known miRNAs and 92 novel miRNAs. Of these, 155 sex-biased DE miRNAs were significantly enriched in sex-associated pathways or GO terms, such as the steroid biosynthesis pathway and reproductive processes. Our integrated miRNA-mRNA analysis demonstrated that sex-biased DE miRNAs and known sexual differentiation-associated genes formed an intricate signaling pathway that regulated sexual differentiation in the yellowfin seabream. Our data provide important molecular resources that help to clarify the roles and mechanisms by which miRNAs affect gonadal development in the yellowfin seabream.

## Figures and Tables

**Figure 1 ijms-21-05690-f001:**
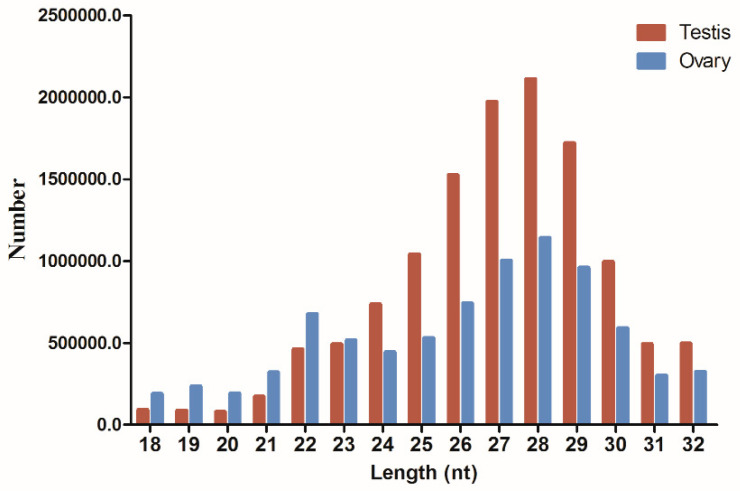
Length distributions of the small RNA sequences in the testis and ovary of the yellowfin seabream.

**Figure 2 ijms-21-05690-f002:**
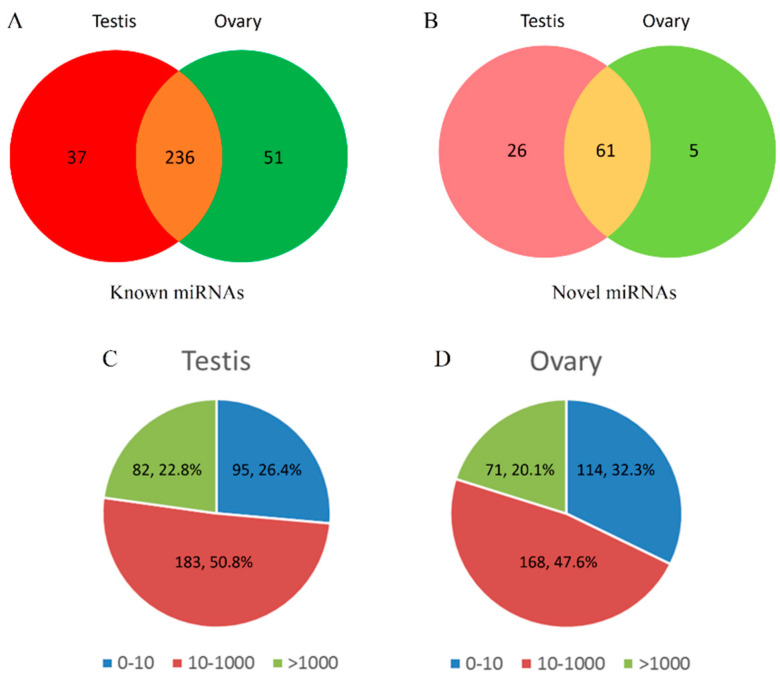
Overview of microRNA (miRNA) expression characteristics. Distributions of known miRNAs (**A**) and novel miRNAs (**B**) between the testis and the ovary. Read distribution intervals and the corresponding proportions in the testis (**C**) and the ovary (**D**).

**Figure 3 ijms-21-05690-f003:**
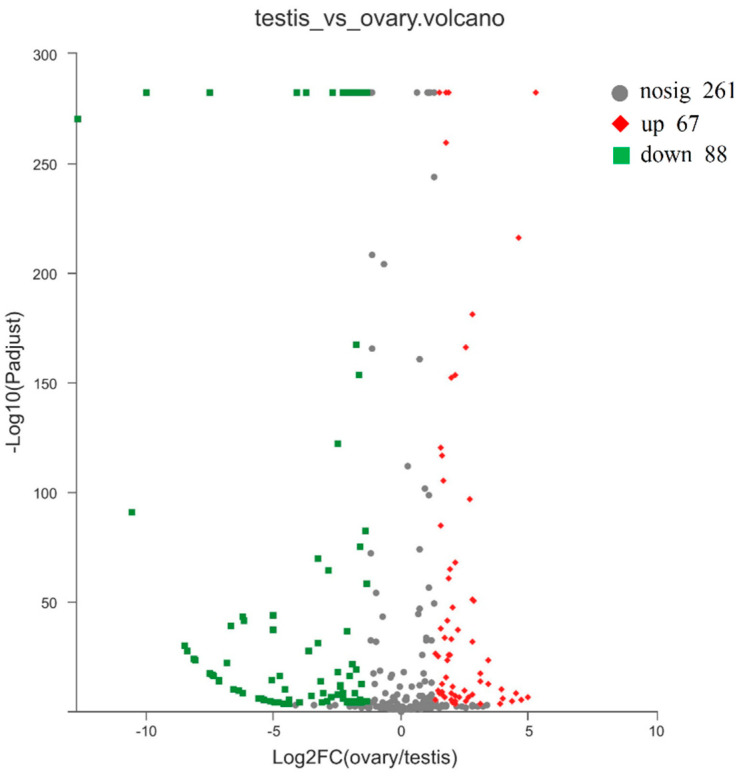
Volcano plot showing the miRNAs differentially expressed between the testis and the ovary. The ovary-biased miRNAs and testis-biased miRNAs are shown as red and green dots, respectively, whereas the miRNAs not differentially expressed are shown as gray dots.

**Figure 4 ijms-21-05690-f004:**
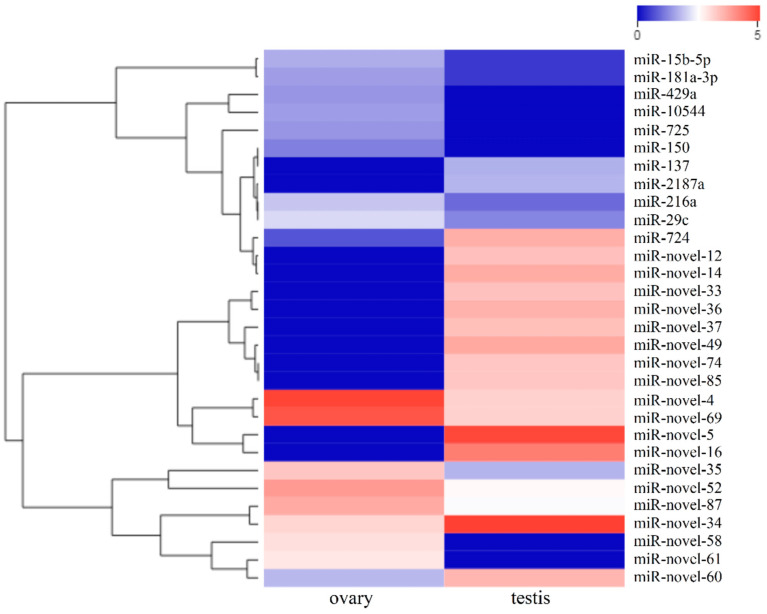
Hierarchical clustering of the 30 most differentially expressed miRNAs between the testis and the ovary. Heatmaps are used to identify differences in expression patterns. Blue, white, and red represent the low-, middle-, and high-frequency miRNAs sequenced in the library, respectively.

**Figure 5 ijms-21-05690-f005:**
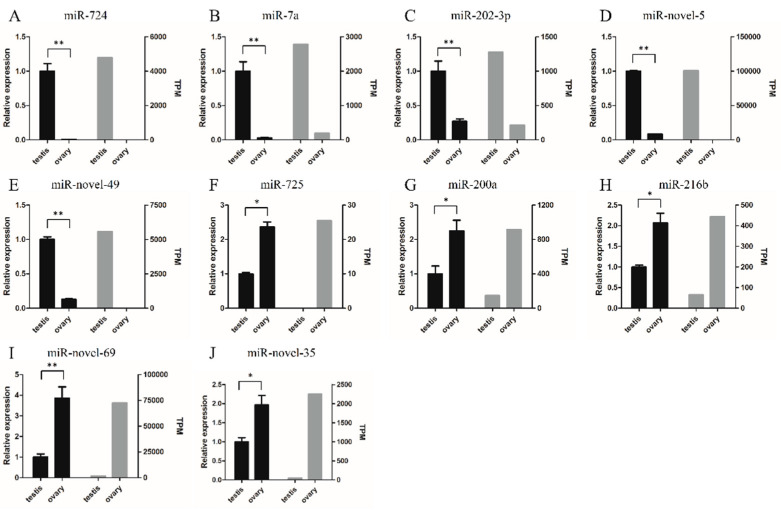
Expression patterns in qRT-PCR (left) and RNA-seq (right). Student’s *t* test, * *p* < 0.05 and ** *p* < 0.01.

**Figure 6 ijms-21-05690-f006:**
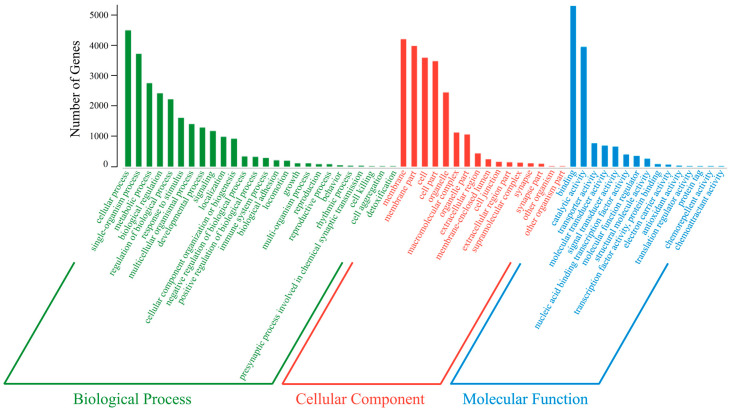
The Gene Ontology annotations of the predicted target genes of the DE miRNAs. The 57 subcategories are plotted on the x-axis, while the number of genes in a specific functional cluster are shown on the y-axis.

**Figure 7 ijms-21-05690-f007:**
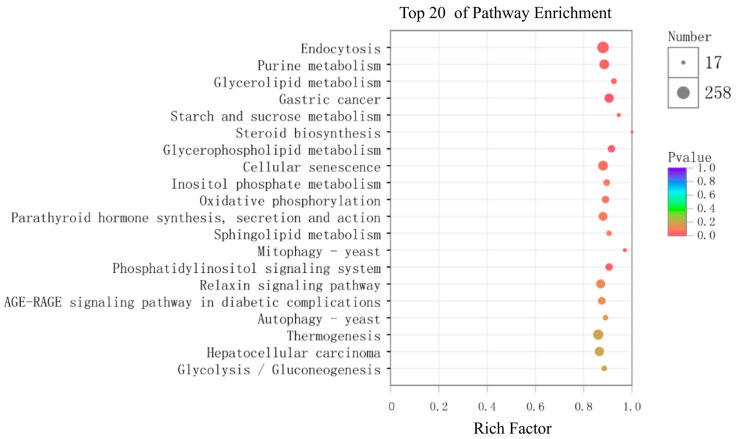
Bubble plot showing the 20 most enriched KEGG (Kyoto Encyclopedia of Genes and Genomes) pathways. The x-axis shows the rich factor, while the y-axis indicates the KEGG pathway terms.

**Figure 8 ijms-21-05690-f008:**
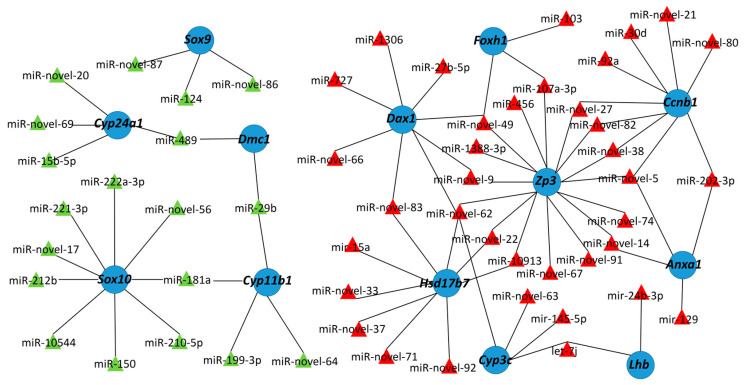
Regulatory network showing interactions between sex-related target genes and DE miRNAs. The testis- and ovary-biased miRNAs are represented by red and green triangles, respectively, while the sex-related target genes are represented by blue circles.

**Table 1 ijms-21-05690-t001:** The top 10 abundantly expressed miRNAs of the two libraries.

miRNA Name	Testis (TPM)	miRNA Name	Ovary (TPM)
miR-novel-67	165,540.54	miR-novel-70	251,216.62
miR-novel-70	143,232.52	miR-novel-47	186,528.19
miR-novel-34	126,233.38	miR-novel-4	113,768.55
miR-novel-5	100,975.98	miR-novel-69	72,700.3
miR-30d	88,730.46	miR-143	68,346.75
miR-novel-29	66,495.07	miR-novel-44	48,189.91
miR-novel-47	60,810.81	miR-30d	39,119.79
miR-143	46,522.26	miR-181c	30,041.2
miR-novel-46	31,960.53	miR-novel-75	29,495.55
miR-15b	25,366.58	miR-novel-46	24,154.3

The highly abundant miRNAs in both testis and ovary are highlighted in red.

**Table 2 ijms-21-05690-t002:** Nine important tests- or ovary-biased miRNAs in yellowfin seabream and six reported fishes.

Species Name	let-7	miR-10	miR-7	miR-9	miR-202	miR-200	miR-29	miR-21	miR-725
*Acanthopagrus latus*	T	T	T	T	T	O	O	O	O
*Pelteobagrus fulvidraco*	T	-	T	T	O	T	T	O	-
*Oreochromis niloticus*	-	T	T	O	O	T	O	O	O
*Oplegnathus punctatus*	O	T	-	-	T	O	-	T	-
*Acipenser schrenckii*	T	T	-	T	-	T	O	T	T
*Trachinotus ovatus*	-	O	T	T	T	-	T	-	-
*Odontobutis potamophila*	T	-	-	-	T	-	T	O	T

Notes: T, testis-biased miRNA; O, ovary-biased miRNA.

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
