# Peer review of "Identification and Comparison of microRNAs in the Gonad of the Yellowfin Seabream (Acanthopagrus Latus)"

_ijms, 2020, doi:10.3390/ijms21165690_

Round 1

Reviewer 1 Report

This is an important article concerning the characterization of the Micro-RNAs of the mature gonads in a commercially relevant fish species, the yellowfin seabream (Acanthopagrus latus), which have been shown to play crucial roles in gametogenesis and gonadal development in vertebrates. It profiles the expression patterns of sex-biased miRNAs in this species, providing important molecular resources that may help to clarify the miRNA-mediated post-transcriptional regulation of sexual differentiation and gonadal development.

More specifically, the authors generated two libraries of small RNAs, constructed from the testes and ovaries of yellowfin seabream that, has far as I understood, were generated from samples collected from 3 specimens with 1-year old and from 3 specimens with 3-years old, being the first ones in the “male stage” and the second ones in the “female stage”. This is my first criticism: it is not “crystal clear” for the readers, not familiarized with this species the exact reason why they choose these two time-points. I interpreted that it was to obtain testis and later ovaries, but this is not clearly stated.

Across both gonads, they identified 324 conserved miRNAs and 92 novel miRNAs: 67 ovary-biased, including the miR-200 families, the miR-29 families, miR-21, and miR-725; and 88 testis-biased, including the let-7 families, the miR-10 families, miR-7, miR-9, and miR-202-3p. GO annotations and KEGG enrichment analyses of putative target genes indicated that many target genes were significantly enriched in the steroid biosynthesis pathway and in the reproductive process, reinforcing the idea that Micro-RNAs are essential for sex determination. In addition, their integrated miRNA-mRNA analysis demonstrated a putative negatively correlated expression pattern in yellowfin seabream gonads.

More specific comments:

Introduction

  • “Yellowfin seabream (Acanthopagrus latus), a fish found in Asian coastal waters, is commercially important because of its beautiful appearance, delicious taste, and high nutritional value”

Comment: the sentence should be straight to the point, I suggest: ”…because of its appearance, taste and nutritional value”.

Results

  • Figures 6 and 7 may benefit from the homogenization of the font type and size (if possible matching the font of the main text). In figure 6 the legend of each graphic bar loses resolution when amplified.
  • “These results suggested that sexual differentiation and gonadal development might be regulated by a complicated miRNA-mRNA network in yellowfin seabream.”

Comment: I suggest the replacement of “complicated” by “complex” or “intricate”.

Discussion

  • “Recently, miRNAs have been recognized as key post-transcriptional regulators of gene expression in many biological processes [8-13]”.

Comment: If the authors refer to literature from 2014, is not that recent. Furthermore, the selection of articles cited here is somehow arbitrary. They should cite a good review of the function of miRNAs as key post-transcriptional regulators or the original article with this finding.

  • “Small RNA high-throughput sequencing has identified sex-related miRNAs in a wide range of fish species, including Pelteobagrus fulvidraco [18, 19], Oncorhynchus mykiss [20, 21], Oplegnathus punctatus [22], Acipenser schrenckii [25], Trachinotus ovatus [26], and Odontobutis potamophila [27].”

Comment: The authors should also cite zebrafish work here, given the relevance of the species as a model organism.

  • It would be interesting to present a figure comparing the results between different species of fish. Moreover, the text should be built in order to compare the results obtained in Yellowfin seabream with the results in other species and with the functions attributed to the miRNAs.

Materials and Methods

  • Sample collection: it is unclear the sampling, are the three 1-year-olds all males and the three 3-year-olds all female, is that it?

Author Response

Reviewer 1

This is an important article concerning the characterization of the Micro-RNAs of the mature gonads in a commercially relevant fish species, the yellowfin seabream (Acanthopagrus latus), which have been shown to play crucial roles in gametogenesis and gonadal development invertebrates. It profiles the expression patterns of sex-biased miRNAs in this species, providing important molecular resources that may help to clarify the miRNA-mediated post-transcriptional regulation of sexual differentiation and gonadal development.

More specifically, the authors generated two libraries of small RNAs, constructed from the testes and ovaries of yellowfin seabream that, has far as I understood, were generated from samples collected from 3 specimens with 1-year old and from 3 specimens with 3-years old, being the first ones in the “male stage” and the second ones in the “female stage”. This is my first criticism: it is not “crystal clear” for the readers, not familiarized with this species the exact reason why they choose these two time-points. I interpreted that it was to obtain testis and later ovaries, but this is not clearly stated.

Across both gonads, they identified 324 conserved miRNAs and 92 novel miRNAs: 67 ovary-biased, including the miR-200 families, the miR-29 families, miR-21, and miR-725; and 88 testis-biased, including the let-7 families, the miR-10 families, miR-7, miR-9, and miR-202-3p. GO annotations and KEGG enrichment analyses of putative target genes indicated that many target genes were significantly enriched in the steroid biosynthesis pathway and in the reproductive process, reinforcing the idea that Micro-RNAs are essential for sex determination. In addition, their integrated miRNA-mRNA analysis demonstrated a putative negatively correlated expression pattern in yellowfin seabream gonads.

Answer: Thank you for the positive comments. We have corrected the sample information, seen in materials and methods in lines 312-315 “Six yellowfin seabreams were selected: three 1-year-old males, with average body weight (ABW) of 113.3 ± 8.5 g and an average body length (ABL) of 15.3 ± 0.3 cm, and three 3-year-old females, with an ABW of 519.3 ± 55.3 g and an ABL of 26.1 ± 0.8 cm.”.

More specific comments:

  • Introduction

“Yellowfin seabream (Acanthopagrus latus), a fish found in Asian coastal waters, is commercially important because of its beautiful appearance, delicious taste, and high nutritional value”

Comment: the sentence should be straight to the point, I suggest: ”…because of its appearance, taste and nutritional value”.

Answer: Thanks for your recommendation. The sentence has been corrected into “Yellowfin seabream (Acanthopagrus latus), a fish found in Asian coastal waters, is commercially important because of its appearance, taste and nutritional value” (Lines 61-62).

  • Results
  1. Figures 6 and 7 may benefit from the homogenization of the font type and size (if possible matching the font of the main text). In figure 6 the legend of each graphic bar loses resolution when amplified.

Answer: Thanks for your recommendation. We have provided clearer images of figure 6 and figure 7 in this version.

  1. “These results suggested that sexual differentiation and gonadal development might be regulated by a complicated miRNA-mRNA network in yellowfin seabream.”

Comment: I suggest the replacement of “complicated” by “complex” or “intricate”.

Answer: Thanks for your recommendation. The sentence has been corrected into “These results suggested that sexual differentiation and gonadal development might be regulated by an intricate miRNA-mRNA network in yellowfin seabream. (Lines 201-202)”

  • Discussion
  1. “Recently, miRNAs have been recognized as key post-transcriptional regulators of gene expression in many biological processes [8-13]”.

Comment: If the authors refer to literature from 2014, is not that recent. Furthermore, the selection of articles cited here is somehow arbitrary. They should cite a good review of the function of miRNAs as key post-transcriptional regulators or the original article with this finding.

Answer: Thanks for your recommendation. We have provided several new and good reviews (Line 208).

2.“Small RNA high-throughput sequencing has identified sex-related miRNAs in a wide range of fish species, including Pelteobagrus fulvidraco [18, 19], Oncorhynchus mykiss [20, 21], Oplegnathus punctatus [22], Acipenser schrenckii [25], Trachinotus ovatus [26], and Odontobutis potamophila [27].”

Comment: The authors should also cite zebrafish work here, given the relevance of the species as a model organism.

Answer: Thanks for your recommendation. We have added the references to the sex-related miRNAs in zebrafish (Line 211).

  1. It would be interesting to present a figure comparing the results between different species of fish. Moreover, the text should be built in order to compare the results obtained in Yellowfin seabream with the results in other species and with the functions attributed to the miRNAs.

Answer: Thanks for your recommendation. We have summarized nine important testis-or ovary-biased miRNAs in yellowfin seabream and six reported fishes (Table 2). The expression profiles of these miRNAs were compared between yellowfin seabream and different species of fish (Lines 222-234).

  • Materials and Methods

Sample collection: it is unclear the sampling, are the three 1-year-olds all males and the three 3-year-olds all female, is that it?

Answer: Yes, the testes and ovaries were collected from one-year-old males and three-year-old females, respectively. Yellowfin seabream, a protandrous hermaphroditic fish, develops as a functional testis during the first reproductive cycle and undergoes a long term of sex reversal to develop a functional ovary at the third year [1].

[1] Li S, Lin G, Fang W, et al. Gonadal transcriptome analysis of sex-related genes in the protandrous yellowfin seabream (Acanthopagrus latus). Frontiers in Genetics, 2020, 11: 709. doi: 10.3389/fgene.2020.00709.

Reviewer 2 Report

Shizhu Li et al., submitted a manuscript in International Journal of Molecular Sciences - ID: 871016 entitled: “Identification and comparison of microRNAs in the gonad of the yellowfin seabream (Acanthopagrus latus)”.

The manuscript could be of helpful to understand the biological role of microRNAs in gonadal development and differentiation in seabream (Acanthopagrus latus) considering that in the last years, massive transcriptomic analysis revealed a large number of microRNAs involved in gametogenesis. Here, the authors, reporting the identification of many new microRNAs specific for testis and ovary in seabream, provided new insight in the scenario of complex role of microRNAs adding new target genes that regulate the miRNA-mRNA network.

I believe that the MS would be suitable for publication in Journal of Molecular Sciences after the minor point suggested.

Minor point to be improved:

in Material and Methods

4.3 RNA extraction…. Add more details concerning it.

4.6 Real time …. Add major experimental details.

Author Response

Reviewer 2

    The manuscript could be of helpful to understand the biological role of microRNAs in gonadal development and differentiation in seabream (Acanthopagrus latus) considering that in the last years, massive transcriptomic analysis revealed a large number of microRNAs involved in gametogenesis. Here, the authors, reporting the identification of many new microRNAs specific for testis and ovary in seabream, provided new insight in the scenario of complex role of microRNAs adding new target genes that regulate the miRNA-mRNA network.

I believe that the MS would be suitable for publication in Journal of Molecular Sciences after the minor point suggested.

Answer: Thank you for the positive comments.

Minor point to be improved:

in Material and Methods

4.3 RNA extraction…. Add more details concerning it.

Answer: Thanks for your recommendation. We have added more experimental details in M&M 4.3 (Lines 319-331).

4.6 Real time …. Add major experimental details.

Answer: Thanks for your recommendation. We have added more experimental details in M&M 4.6 (Lines 353-366).